# Shifts: A Dataset of Real Distributional Shift Across Multiple Large-Scale Tasks

**Andrey Malinin**[1,2]       Neil Band[5]       Yarin Gal[5,6]       Mark J. F. Gales[4]

Alexander Ganshin [1]       German Chesnokov[1]       Alexey Noskov[1]

Andrey Ploskonosov[1]       Liudmila Prokhorenkova[1,2,3]       Ivan Provilkov[1,3]

Vatsal Raina[4]       Vyas Raina[4]       Denis Roginskiy [1]       Mariya Shmatova[1]

Panos Tigas[5]       Boris Yangel[1]

am969@yandex-team.ru

## Abstract

There has been significant research done on developing methods for improving robustness to distributional shift and uncertainty estimation. In contrast, only limited work has examined developing standard datasets and benchmarks for assessing these approaches. Additionally, most work on uncertainty estimation and robustness has developed new techniques based on small-scale regression or image classification tasks. However, many tasks of practical interest have different modalities, such as tabular data, audio, text, or sensor data, which offer significant challenges involving regression and discrete or continuous structured prediction. Thus, given the current state of the field, a standardized large-scale dataset of tasks across a range of modalities affected by distributional shifts is necessary. This will enable researchers to meaningfully evaluate the plethora of recently developed uncertainty quantification methods, as well as assessment criteria and state-of-the-art baselines. In this work, we propose the *Shifts Dataset* for evaluation of uncertainty estimates and robustness to distributional shift. The dataset, which has been collected from industrial sources and services, is composed of three tasks, with each corresponding to a particular data modality: tabular weather prediction, machine translation, and self-driving car (SDC) vehicle motion prediction. All of these data modalities and tasks are affected by real, "in-the-wild" distributional shifts and pose interesting challenges with respect to uncertainty estimation. In this work we provide a description of the dataset and baseline results for all tasks.

## 1 Introduction

Machine learning models are being applied to numerous areas [1, 2, 3, 4, 5, 6] and are widely deployed in production. An assumption which pervades all of machine learning is that the training, validation, and deployment data are independent and identically distributed (i.i.d.). Thus, good performance and generalization on validation data imply that the model will perform well in deployment. Unfortunately, this assumption seldom holds in real, "in the wild", applications. In practice, data are subject to a wide range of possible *distributional shifts* — mismatches between the training data, and test or deployment data [7, 8, 9]. In general, the greater the degree of shift, the poorer is the model's performance. The problem of distributional shift is of relevance not only to academic researchers, but to the machine learning community at-large. Indeed, *all* ML practitioners have faced the issue of

---

[1]Yandex, [2]HSE University, [3]Moscow Institute of Physics and Technology, [4]ALTA Institute, University of Cambridge, [5]University of Oxford, [6]Alan Turing Institute

mismatch between the training and test sets. This is especially important in high-risk applications of machine learning, such as finance, medicine, and autonomous vehicles. Indeed, in such applications a mistake on part of an ML system may incur financial or reputational loss, or possible loss of life. It is therefore increasingly important to assess both a model's *robustness* to distribution shift and its estimates of *predictive uncertainty*, which enable it to detect distributional shifts [10, 11, 12].

The area of uncertainty estimation and robustness has developed rapidly in recent years. Model averaging [13, 14, 15, 16] has emerged as the de-facto standard approach to uncertainty estimation. Ensemble- and sampling-based uncertainty estimates have been successfully applied in detecting misclassifications, out-of-distribution inputs, adversarial attacks [17, 18], and for active learning [19]. Recently, such approaches have been extended to structured prediction tasks such as machine translation and speech recognition [20, 21, 22, 23, 24]. However, these approaches require large computational and memory budgets. Works using temperature scaling [25, 26] and other recent approaches in deterministic uncertainty estimation [27, 28, 29, 30, 31] aim to tackle this issue, but have only recently become comparable to ensemble methods [30, 31]. Prior Networks [32, 33, 34] — models which *emulate* the mechanics of an ensemble — have been proposed as a deterministic single model approach to uncertainty estimation which are competitive with ensembles. However, they require distributionally shifted training data, which may not be feasible in many applications. Prior Networks have also been used for *Ensemble Distribution Distillation* [35, 34, 36] — a distillation approach through which the predictive performance and uncertainty estimates of an ensemble are captured within a single Prior Network, reducing the inference cost to that of a single model.

While much work has been done on developing *methods*, limited work has focused on new datasets and benchmarks. In [37, 38], the authors introduced benchmarks for uncertainty quantification in Bayesian deep learning but only considered the image-based task of classifying diabetic retinopathy. Recently, a range of works by Hendrycks et al. [39, 40, 41] proposed a set of datasets based on ImageNet [42] for evaluating model robustness to various types of distributional shifts. These datasets — ImageNet C, A, R, and O — include synthetically added noise, natural adversarial attacks, renderings, and previously unseen classes of objects.[1] The release of WILDS, a collection of datasets containing real-world distributional shifts [8], similarly represents a significant step forward, but again mostly focuses on images. Finally, the MTNT dataset [43], which contains many examples of highly atypical usage of language, such as acronyms, profanity, emojis, slang, and code-switching, has been used at the Workshop on Machine Translation (WMT) robustness track. However, it has not been considered by the uncertainty community in the context of *detecting* distributional shift.

Unfortunately, with a few exceptions, most work on uncertainty estimation and robustness has focused on developing new methods on small-scale tabular regression or image classification tasks, such as UCI, MNIST [44], Omniglot [45], SVHN [46], and CIFAR10/100 [47]. Few works have been evaluated on the ImageNet variations A, R, C, and O, or WILDS. However, even evaluation on these datasets is limited, as they mainly focus on classification of images, and sometimes text. In contrast, many tasks of practical interest have different modalities, such as tabular data (in medicine and finance), audio, text, or sensor data. Furthermore, these tasks are not always classification; they often involve regression and discrete or continuous structured prediction. Given the current state of the field, we aim to draw the attention of the community to the evaluation of uncertainty estimation and robustness to distributional shift on a realistic set of large-scale tasks across a range of modalities. This is necessary to meaningfully evaluate the plethora of methods for uncertainty quantification and improved robustness, and to accelerate the development of this area and safe ML in general.

In this work, we propose the **Shifts Dataset**[2] for evaluation of uncertainty estimates and robustness to distributional shift. This dataset consists of data taken directly from large-scale industrial sources and services where distributional shift is ubiquitous — settings as close to "in the wild" as possible. The dataset is composed of three parts, with each corresponding to a particular data modality: *tabular weather prediction* data provided by the Yandex Weather service; *machine translation* data taken from the WMT robustness track and mined from Reddit, and annotated in-house by Yandex Translate; and, self-driving car (SDC) data provided by Yandex SDG, for the task of *vehicle motion prediction*. All of these data modalities and tasks are affected by distributional shift and pose interesting challenges with respect to uncertainty estimation. This paper provides a detailed analysis of the data as well as baseline uncertainty estimation and robustness results using ensemble methods.

---

[1] ImageNet has only "natural" images; thus, renderings represent a shift in texture, but not content.
[2] Data and example code are available at `https://github.com/yandex-research/shifts`

## 2   Evaluation Paradigm, Metrics, and Baselines

**Paradigm**   In most prior work, uncertainty estimation and robustness have been assessed separately. Robustness to distributional shift is typically assessed via metrics of predictive performance on a particular task — given two (or more) evaluation sets, where one is considered matched to the training data and the other(s) shifted, models which have a smaller degradation in performance on the shifted data are considered more robust. The quality of uncertainty estimates is often assessed via the ability to classify whether an example came from the "in-domain" dataset or a shifted dataset using measures of uncertainty. Here, performance is assessed via Area under a Receiver-Operator Curve (ROC-AUC %) or Precision-Recall curve (AUPR %). While these evaluation paradigms are meaningful, we believe that they are two halves of a common whole. Instead, we consider the following paradigm:

*As the degree of distributional shift increases, so does the likelihood that a model makes an error and the degree of this error. Models should yield uncertainty estimates which correlate with the degree of distributional shift, and therefore are indicative of the likelihood and the degree of the error.*

This paradigm is more general, as a model may be robust to certain examples of distributional shift and yield accurate, low uncertainty predictions. A model may also perform poorly and yield high estimates of uncertainty on underrepresented data matched to the training set. Thus, splitting a dataset into "in-domain" and "out-of-distribution" may not yield partitions on which a model strictly performs well or poorly, respectively. Instead, it is necessary to *jointly* assess robustness and uncertainty estimation, in order to see whether uncertainty estimates at the level of a single prediction correlate well with the likelihood or degree of error. Thus, we view the problems of robustness and uncertainty estimation as having *equal* importance — models should be robust, but where they are not, they should yield high estimates of uncertainty, which enables risk-mitigating actions to be taken (e.g., transferring control of a self-driving vehicle to a human operator).

Additionally, we assume that at training or test time *we do not know a priori* about alternative domains and whether or how our data is shifted. This setup aims to emulate real-world deployments in which the variation of conditions is vast and one can never collect enough data to cover all situations. It is for this reason we view robustness and uncertainty as equally important — we assume that one can never be fully robust in all situations, and it is in these situations that high-quality uncertainty estimation is crucial. This is a strictly more challenging setting than one in which auxiliary information about the degree or nature of shift is available at training or test time (e.g., in WILDS [8]).

We have constructed the Shifts Dataset within the context of this paradigm. Specifically, the dataset is constructed with the following attributes. First, the annotations of distributional shift are meant to be used for analysis rather than model construction. Second, we have "canonically" partitioned the datasets such that the shifts are realistic but significant and to which it is challenging to be fully robust — this allows us to assess the quality of uncertainty estimates. However, the weather and motion prediction datasets *can* be repartitioned in alternative ways which are different from our canonical partitioning, such that alternative robustness paradigms can be evaluated.[3]

**Assessment Metrics**   We jointly assess robustness and uncertainty via *error-retention curves* [12, 14] and *F1-retention curves*. Given an error metric, such as MSE or GLEU, error-retention curves trace the error over a dataset as a model's predictions are replaced by ground-truth labels in order of decreasing uncertainty. F1-retention curves depict the F1 for predicting whether a model's predictions are sufficiently good based on uncertainties (here we vary retention fraction, i.e., the fraction of data with the smallest uncertainty values that we classify as acceptable). Both assess the performance of a hybrid human-AI system, where a model can consult an oracle (human) for assistance in difficult situations. The area under this curve can be decreased (error retention) or increased (F1 retention) either by improving the predictive performance of the model, such that it has lower overall error, or by providing better estimates of uncertainty, such that more errorful predictions are rejected earlier. Thus, the area under the error (R-AUC) and F1 (F1-AUC) retention curves are metrics which jointly assess robustness to distributional shift and the quality of uncertainty estimates. We also quote F1 at 95% retention rate. These metrics, detailed in Appendix A, are used for all tasks in this paper.

**Choice of Baselines**   In this work we consider ensemble-based baselines. This was done for a number of reasons. First, ensemble-based approaches are a standard way to obtain *both* improved

---

[3]Tools for partitioning and repartitioning are provided in our GitHub repository.

robustness versus single models *and* interpretable uncertainty estimates. Ensembles improve robustness because each model represents a functionally different explanation of the data. Thus, even if each individual model in an ensemble is subject to spurious correlations, the models will have different spurious correlations. When the models are combined, the effects of spurious correlations are cancelled out to a certain degree, improving generalization performance. Second, ensemble methods are straightforward to apply to any task of choice and require little adaptation. Uncertainty estimates can be obtained from measures of ensemble diversity — if the predictions are highly diverse, then the ensemble members cannot agree on what the prediction should be and therefore are highly uncertain. Other than ensemble methods, there are few alternative approaches which are known to yield improved robustness *and* interpretable uncertainty estimates, can be easily applied to a broad range of large-scale tasks without significant adaptation, and do not require information about the nature of distributional shift at training or test time. We leave the exploration of these alternatives and the development of new ones to future work. We do not examine robust learning methods, such as IRM [48, 8], as they require domain annotations at training time and do not yield uncertainty estimates.

## 3    Tabular Weather Prediction

Uncertainty estimation and robustness are essential in applications like medical diagnostics and financial forecasting. In such applications, data is often represented in a heterogeneous tabular form. While it is challenging to obtain either a large medical or financial dataset, the Yandex Weather service has provided a large tabular Weather Prediction dataset that features a natural tendency for the data distribution to drift over time (concept drift [49, 50]). Furthermore, the locations are non-uniformly distributed around the globe based on population density, land coverage, and observation network development, which means that certain climate zones, like the Polar regions or the Sahara, are under-represented. We argue that this tabular Weather Prediction data represents similar challenges to the ones faced on financial and medical data, which is often combined from different hospitals/labs, consists of population-groups that are non-uniformly represented, and has a tendency to drift over time. Thus, the data we consider in this paper can be used as an appropriate benchmark for developing more robust models and uncertainty estimation methods for tabular data.

**Dataset**    The Shifts Weather Prediction dataset contains a scalar regression and a multi-class classification tasks: at a particular latitude, longitude, and timestamp, one must predict either the air temperature at two meters above the ground or the precipitation class, given targets and features derived from weather station measurements and weather forecast models. The data consists of 10 million 129-column entries: 123 meteorological features, 4 meta-data attributes (time, latitude, longitude and climate type) and 2 targets — temperature (target for regression task) and precipitation class (target for classification task). The full feature list is provided in Section C.2. It is important to note that the features are highly heterogeneous, i.e., they are of different types and scales. The full data is distributed uniformly between September $1^{st}$, 2018, and September $1^{st}$, 2019, with samples across all climate types. This data is used by Yandex for real-time weather forecasts and represents a real industrial application.

To provide a standard benchmark that contains both in-domain and shifted data, we use a particular "canonical partitioning"[4] of the full dataset into training, development (dev), and evaluation (eval) datasets. The training, in-domain dev (dev_in) and in-domain eval (eval_in) data consist of measurements made from September 2018 till April $8^{th}$, 2019 for climate types *Tropical*, *Dry*, and *Mild Temperate*. The shifted dev (dev_out) data consists of measurements made from $8^{th}$ July till $1^{st}$ September 2019 for the climate type *Snow*. 50K data points are sub-sampled for the climate type *Snow* within this time range to construct dev_out. The shifted eval data is further shifted than the out-of-domain development data; measurements are taken from $14^{th}$ May till $8^{th}$ July 2019, which is more distant in terms of the time of the year from the in-domain data compared to the out-of-domain development data. The climate types are restricted to *Snow* and *Polar*. Further details are provided in Appendix C.1. Details on use and support plan are in Appendix B.

---

[4]Alternative partitionings can be made from the full data, but we use the canonical partitioning throughout this work, and also for the Shifts Challenge: http://research.yandex.com/shifts

**Baselines** To build baseline models for the temperature prediction and precipitation classification tasks, we use the open-source CatBoost gradient boosting library that is known to achieve state-of-the-art results on tabular datasets [51]. We use an ensemble-based approach to uncertainty estimation for GBDT models [52]. For each task, an ensemble of ten models is trained on the training data with different random seeds. For regression, the models predict the mean and variance of the normal distribution by optimizing the negative log-likelihood. For classification, the models predict a probability distribution over precipitation classes. Training details are provided in Appendix C.4. Additional ensemble-based baselines and results are provided in Appendix C.5.

We first compare the predictive performance of ensembles and single models; the results are shown in Table 1. Firstly, we observe that all models perform worse on shifted data than on in-domain data. For regression, we observe that the RMSE of the ensemble (on the `eval` set) is about two degrees Celsius. Note that ensembling allows us to reduce RMSE by about $0.16°$ compared to a single model. Similarly, ensembling reduces the MAE by approximately $0.12°$. For classification, ensembling boosts the accuracy and macro-averaged F1 by about 2%. Note that the classification task is unbalanced (see Appendix C.1 for details), so for better interpretability, we also report the accuracy and Macro-F1 of the classifier always predicting the majority class.

Table 1: Predictive performance for Weather Prediction. Mean $\pm \sigma$ is quoted for the single models.

| Data | Regression | | | | Classification | | | | | |
|---|---|---|---|---|---|---|---|---|---|---|
| | RMSE ↓ | | MAE ↓ | | Accuracy (%) ↑ | | | Macro F1 (%) ↑ | | |
| | Single | Ens | Single | Ens | Maj. | Single | Ens | Maj. | Single | Ens |
| dev-in | $1.59_{\pm 0.00}$ | 1.51 | $1.18_{\pm 0.00}$ | 1.11 | 37.9 | $67.0_{\pm 0.075}$ | 68.5 | 17.2 | $48.8_{\pm 0.8}$ | 51.0 |
| dev-out | $2.30_{\pm 0.01}$ | 2.12 | $1.75_{\pm 0.01}$ | 1.61 | 35.7 | $47.5_{\pm 0.249}$ | 50.3 | 19.4 | $27.9_{\pm 0.5}$ | 29.0 |
| dev | $1.98_{\pm 0.01}$ | 1.84 | $1.47_{\pm 0.01}$ | 1.36 | 36.8 | $57.2_{\pm 0.117}$ | 59.4 | 17.2 | $43.3_{\pm 0.8}$ | 45.6 |
| eval-in | $1.60_{\pm 0.00}$ | 1.52 | $1.19_{\pm 0.00}$ | 1.11 | 37.9 | $66.7_{\pm 0.060}$ | 68.2 | 17.2 | $47.5_{\pm 0.3}$ | 49.8 |
| eval-out | $2.60_{\pm 0.03}$ | 2.37 | $1.91_{\pm 0.01}$ | 1.75 | 30.0 | $44.5_{\pm 0.184}$ | 46.7 | 17.4 | $27.4_{\pm 0.2}$ | 28.8 |
| eval | $2.16_{\pm 0.01}$ | 2.00 | $1.56_{\pm 0.01}$ | 1.44 | 33.9 | $55.5_{\pm 0.090}$ | 57.3 | 17.4 | $39.3_{\pm 0.2}$ | 41.4 |

Next, we jointly evaluate the robustness and uncertainty estimates for ensembles and single models. For the regression task, we use the predicted variance as the uncertainty measure of a single model. For ensembles, we use the total variance (tvar) that is the sum of the variance of the predicted mean and the mean of the predicted variance [11, 12, 52]. For the classification task, we use the entropy of the prediction as the uncertainty measure of a single model. For ensembles, we use the (negated) confidence. We measure the area under the error-retention and F1-retention curves as described in Appendices A and C.3. These two performance metrics are denoted as R-AUC and F1-AUC, respectively. A good uncertainty measure is expected to achieve low R-AUC and high F1-AUC. Additionally, we report the F1 score at a retention rate of 95% of the most certain samples (F1@95%). All these measures jointly assess the predictive performance and quality of uncertainty estimates.

Table 2: Retention performance for Weather Prediction. Mean $\pm \sigma$ is quoted for the single models.

| | Data | R-AUC ↓ | | F1-AUC (%) ↑ | | F1@95% ↑ | |
|---|---|---|---|---|---|---|---|
| | | Single | Ens | Single | Ens | Single | Ens |
| Regression | dev | $1.894_{\pm 0.017}$ | 1.227 | $44.35_{\pm 0.2}$ | 52.20 | $62.72_{\pm 0.1}$ | 65.83 |
| | eval | $2.320_{\pm 0.063}$ | 1.335 | $43.41_{\pm 0.1}$ | 52.36 | $61.89_{\pm 0.1}$ | 64.72 |
| Classification | dev | $0.1666_{\pm 0.001}$ | 0.1522 | $57.72_{\pm 0.1}$ | 59.07 | $73.04_{\pm 0.1}$ | 74.86 |
| | eval | $0.1799_{\pm 0.001}$ | 0.1640 | $56.25_{\pm 0.1}$ | 58.22 | $71.56_{\pm 0.1}$ | 73.17 |

The results are shown in Table 2. Here, as expected, ensembles significantly outperform single models. This observation is consistent over all considered evaluation measures. The associated retention curves are provided in Figure 1 for `eval` and Figure 11 in Appendix C for `dev`.

Finally, we conduct a comparison of different uncertainty measures. For this, we measure F1-AUC discussed above and ROC-AUC that evaluates uncertainty-based out-of-distribution (OOD) data detection. The results are shown in Table 3. In this experiment, we do not evaluate single models. For regression, we consider the following uncertainty measures: total variance (tvar) discussed above

Table 3: Comparing uncertainty measures of CatBoost ensembles for Weather Prediction.

| Data | | Regression | | | Classification | | | | |
|---|---|---|---|---|---|---|---|---|---|
| | | Total Unc. | Knowledge Unc. | | Total Unc. | | Knowledge Unc. | | |
| | | tvar | varm | EPKL | Conf | Entropy | MI | EPKL | RMI |
| dev | F1-AUC (%) ↑ | **52.20** | 50.12 | 50.51 | **59.07** | 58.86 | 57.72 | 57.69 | 57.66 |
| | ROC-AUC (%) ↑ | 62.96 | 82.31 | **85.29** | 63.98 | 65.00 | 83.75 | 83.96 | **84.12** |
| eval | F1-AUC (%) ↑ | **52.36** | 49.81 | 50.40 | **58.22** | 57.89 | 56.99 | 56.96 | 56.93 |
| | ROC-AUC (%) ↑ | 65.99 | 78.32 | **79.90** | 66.20 | 66.76 | 83.44 | 83.59 | **83.68** |

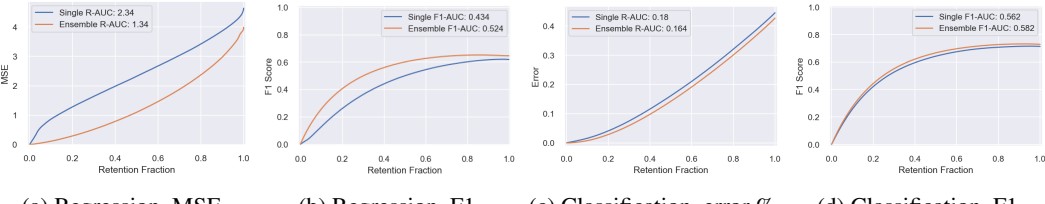

(a) Regression, MSE.  (b) Regression, F1.  (c) Classification, error %.  (d) Classification, F1.

Figure 1: Retention curves with CatBoost on `eval` for the Weather Prediction dataset.

that is a measure of *total uncertainty*, variance of the mean predictions across the ensemble models (varm) and the expected pairwise KL-divergence (EPKL) that are measures of *knowledge uncertainty*. The results show that uncertainty measures that capture knowledge uncertainty perform best at OOD detection, as suggested by the high ROC-AUC values, while the measure of total uncertainty performs best for detecting errors (F1-AUC). Thus, as expected, the choice of a metric to use depends heavily on the task. Among measures of knowledge uncertainty, EPKL has better performance. For classification, the measures of total uncertainty are the negative confidence (Conf) and the entropy of the average prediction (Entropy). The measures of knowledge uncertainty are mutual information (MI), EPKL, and reverse mutual information (RMI). Similar to regression, uncertainty measures that capture knowledge uncertainty are better in terms of ROC-AUC. Among them, reverse mutual information performs best. The measures of total uncertainty are better for F1-AUC, and the best results are achieved with negative confidence.

## 4 Machine Translation

As part of the Shifts Dataset we examine the task of machine translation for the text modality. Translation services, such as Google Translate or Yandex Translate, often encounter atypical and unusual use of language in their translation queries. This typically includes slang, profanities, poor grammar, orthography and punctuation, as well as emojis. This poses a challenge to modern translation systems, which are typically trained on corpora with a more "standard" use of language. Therefore, it is important for models to both be robust to atypical language use to provide high-quality translations, as well as to indicate when they are unable to provide a quality translation.

Translation is inherently a *structured prediction* task, as there are dependencies between the tokens in the output sequence. Often we must make assumptions about the form of these dependencies; for example, most modern translation systems are left-to-right autoregressive. However, we could consider conditionally independent predictions or other factorization orders. The nature of these assumptions makes it challenging to obtain a theoretically sound measure of uncertainty. Only recently has work been done on developing principled uncertainty measures for structured prediction [21, 22, 23, 24, 53]. Nevertheless, this remains an unsolved task and a fruitful area for research.

**Dataset**   The dataset contains training, development (`dev`) and evaluation (`eval`) data, where each set consists of pairs of source and target sentences in English and Russian, respectively. As most production Neural Machine Translation (NMT) systems are built using a variety of general purpose corpora, we use the freely available WMT'20 En-Ru corpus as training data. This dataset primarily focuses on parliamentary and news data that is, for the most part, grammatically and orthographically correct with formal language use. The `dev` and `eval` datasets consist of an "in-domain" partition

matched to the training data, and an "out-of-distribution" or shifted partition, which contains examples of atypical language usage. The in-domain `dev` and `eval` sets are Newstest'19 En-Ru and a newly collected news corpus from GlobalVoices [54], respectively. For the shifted development data we use the Reddit corpus prepared for the WMT'19 robustness challenge [43]. This data contains examples of slang, acronyms, lack of punctuation, poor orthography, concatenations, profanity, and poor grammar, among other forms of atypical language usage. This data is representative of the types of inputs that machine translation services find challenging. As Russian target annotations are not available, we pass the data through a two-stage process, where orthographic, grammatical, and punctuation mistakes are corrected, and the source-side English sentences are translated into Russian by expert in-house Yandex translators. The development set is constructed from the same 1400-sentence test-set used for the WMT'19 robustness challenge. For the evaluation set we use the open-source MTNT crawler which connects to the Reddit API to collect a further set of 3,000 English sentences from Reddit, which is similarly corrected and translated. The shifted `dev` and `eval` data are also annotated with 7 non-exclusive anomaly flags. Details regarding pre-processing, annotations and licenses is available in Appendix D.1. Details on use and support plan are in Appendix B.

**Metrics**   To evaluate the performance of our models we will consider corpus-level BLEU [55] and sentence-level GLEU [56, 57, 58]. As machine translation is a multi-modal task and translation systems often yield multiple translation hypothesis we will consider two GLEU-based metrics for evaluating translation quality. First is the *expected GLEU* or **eGLEU** across all translation hypotheses, where each hypothesis is weighted by a *confidence score*, and confidences across all hypotheses sum to one. Second is the maximum GLEU **maxGLEU** across all hypotheses in the beam. Details of these metrics can be found in Appendix D.2. These metrics are then used to compute the error- and F1-retention curves which jointly assess uncertainty and robustness, as discussed in Appendix A.

**Baselines**   In this work we considered an ensemble baseline based on [24]. Here, we use an ensemble of 3 Transformer-Big [5] models trained on the WMT'20 En-Ru corpus. Models were trained using a fork of FairSeq [59] with a large-batch training set. Beam-Search decoding with a beam-width of 5 is used to obtain translation hypotheses. Hypotheses confidence weights are obtained by exponentiating the negative log-likelihood of each hypothesis and then normalizing across all hypotheses in the beam. Individual models in the ensemble are used as a single-model baseline.

Table 4 presents the predictive performance on the `dev` and `eval` sets as well as on their in-domain and shifted subsets. There is a performance difference of nearly 10 BLEU and GLEU points between the in-domain news and shifted Reddit data, which shows the degradation in quality due to atypical language usage. The ensemble is able to outperform the individual models, which is expected. These results also show that BLEU correlates quite well with eGLEU. maxGLEU shows that significantly better performance is obtainable if we were better at ranking the hypotheses in the beam.

Table 4: Predictive performance for Machine Translation. Mean $\pm \sigma$ is quoted for the single models.

| Data | BLEU ↑ | | eGLEU ↑ | | maxGLEU ↑ | |
| --- | --- | --- | --- | --- | --- | --- |
| | Single | Ens | Single | Ens | Single | Ens |
| `dev-in` | $32.04_{\pm 0.23}$ | 32.73 | $34.45_{\pm 0.10}$ | 35.09 | $41.08_{\pm 0.09}$ | 42.00 |
| `dev-out` | $20.65_{\pm 0.16}$ | 21.06 | $22.66_{\pm 0.07}$ | 23.00 | $28.28_{\pm 0.19}$ | 28.63 |
| `dev` | $28.89_{\pm 0.20}$ | 29.52 | $29.67_{\pm 0.09}$ | 30.19 | $35.89_{\pm 0.12}$ | 36.58 |
| `eval-in` | $29.52_{\pm 0.21}$ | 30.08 | $30.39_{\pm 0.10}$ | 30.88 | $36.19_{\pm 0.19}$ | 36.82 |
| `eval-out` | $21.00_{\pm 0.12}$ | 21.54 | $23.19_{\pm 0.07}$ | 23.60 | $29.35_{\pm 0.11}$ | 29.88 |
| `eval` | $26.39_{\pm 0.17}$ | 26.92 | $26.76_{\pm 0.06}$ | 27.20 | $32.74_{\pm 0.14}$ | 33.31 |

Having evaluated the baselines' predictive performance, we now jointly assess their uncertainty and robustness using the area under the error-retention curve (R-AUC), area under the F1-retention curve (F1-AUC) and F1 at 95% retention, as detailed in Appendices A and D.2. Additionally, we evaluate in terms of % ROC-AUC whether it is possible to discriminate between the in-domain data and the shifted data based on the measures of uncertainty provided by the models. As the measure of uncertainty we use the negative log-likelihood, averaged across all 5 hypotheses. In the case of individual models, this is a measure of *data* or *aleatoric* uncertainty, and in the case of the ensemble, it is a measure of *total uncertainty* [24]. The results show that the ensemble consistently outperforms the single-model baseline.

Table 5: Uncertainty and robustness for Machine Translation. Mean $\pm \sigma$ is quoted for the single models.

| Data | R-AUC $\downarrow$ | | F1-AUC $\uparrow$ | | F1@95% $\uparrow$ | | ROC-AUC (%) $\uparrow$ | |
| | Single | Ens | Single | Ens | Single | Ens | Single | Ens |
| --- | --- | --- | --- | --- | --- | --- | --- | --- |
| dev | $33.22_{\pm 0.48}$ | 32.87 | $0.43_{\pm 0.00}$ | 0.44 | $0.42_{\pm 0.01}$ | 0.43 | $68.90_{\pm 0.28}$ | 69.30 |
| eval | $34.80_{\pm 0.06}$ | 34.57 | $0.37_{\pm 0.07}$ | 0.38 | $0.34_{\pm 0.03}$ | 0.36 | $79.18_{\pm 0.63}$ | 80.10 |

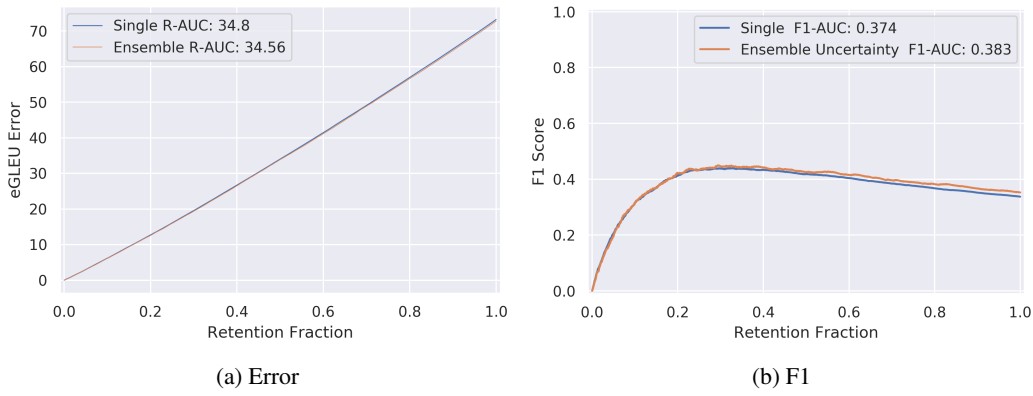

(a) Error        (b) F1

Figure 2: Retention curves using eGLEU on eval data.

# 5   Vehicle Motion Prediction

We present the Shifts Vehicle Motion Prediction dataset to examine the implications of distributional shift in self-driving vehicles. The area of autonomous vehicle (AV) technology is highly relevant for uncertainty and robustness research, as the safety requirements and the risks associated with any errors are high. Furthermore, distributional shift is ubiquitous in the autonomous driving domain. During technology development, most self-driving companies concentrate their fleet in a limited number of locations and routes due to the large cost of operating in a new location. Therefore, fleets often face distributional shift when they begin operation in new locations. It is thus important to transfer as much knowledge as possible from the old locations to new ones. It is also critical for a planning model to recognize when this transferred knowledge is insufficient upon encountering unfamiliar data, which could risk unpredictable and unsafe behavior.[5] Uncertainty quantification therefore has potentially life-critical application in this domain. For example, when the model's predictive uncertainty reaches a particular threshold, the vehicle can exercise extra caution or request assistance from a remote operator.

Motion prediction is among the most important problems in the autonomous driving domain and has recently drawn significant attention from both academia and industry [60, 61, 62, 63, 64, 65, 66, 67, 68, 69, 70, 71]. It involves predicting the distribution over possible future states of agents around the self-driving car at a number of moments in time. A model of possible futures is needed because a self-driving vehicle needs a certain amount of time to change its speed, and sudden changes may be uncomfortable or even dangerous for its passengers. Therefore, in order to ensure a safe and comfortable ride, the motion planning module of a self-driving vehicle must reason about where other agents might end up in a few seconds to avoid planning a potential collision. This problem is complicated by the fact that *the future is inherently uncertain*. For example, we cannot know the high-level navigational goals of other agents, or even their low-level tendency to turn right or left at a T-junction if they fail to indicate one way or another.[6] In order for the planning module to make the right decision, this uncertainty must be precisely quantified. Finally, motion prediction is also interesting because the predictions are both *structured and continuous*. This poses further challenges in uncertainty estimation. Recently, ensemble-based uncertainty estimation for the related task of autonomous vehicle *planning* was examined [72], where a variance-based measure was proposed.

---

[5] A case of knowledge, or epistemic uncertainty [11, 12].

[6] A case of data, or aleatoric uncertainty.

However, there is still much potential for further development of informative measures of uncertainty in continuous structured prediction tasks such as motion prediction.

**Dataset** The dataset for the Vehicle Motion Prediction task was collected by the Yandex Self-Driving Group (SDG) fleet and is the largest vehicle motion prediction dataset released to date, containing 600,000 scenes. These scenes span six locations, three seasons, three times of day, and four weather conditions. Each scene includes information about the state of dynamic objects and an HD map. Each scene is 10 seconds long and is divided into 5 seconds of context features and 5 seconds of ground truth targets for prediction, separated by the time $T = 0$. The goal is to predict the movement trajectory of vehicles at time $T \in (0, 5]$ based on the information available for time $T \in [-5, 0]$. The data contains training, development (`dev`) and evaluation (`eval`) sets. In order to study the effects of distributional shift, we partition the data such that the `dev` and `eval` sets have *in-domain* partitions which match the location and precipitation type of the training set, and *out-of-domain* or *shifted* partitions which do not match the training data along one or more of those axes. As in the other Shifts tasks, we define a canonical partitioning which is used throughout benchmarking.[7] The training set and in-domain partition of the `dev` and `eval` sets are taken from Moscow. Distributionally shifted `dev` data is taken from Skolkovo, Modiin, and Innopolis. Distributionally shifted `eval` data is taken from Tel Aviv and Ann Arbor. We also remove all cases of precipitation from the in-domain sets, while distributionally shifted datasets include precipitation. A full description of the dataset is available in Appendix E, the support plan is detailed in Appendix B.

**Metrics** Here we consider five different performance metrics — minimum Average Displacement Error (minADE), minimum Final Displacement Error (minFDE), confidence-weighed ADE and FDE, and corrected Negative Log-Likelihood (cNLL). cNLL is a new metric we introduce that is particularly well-suited for assessing how models handle multi-modal situations. The minimum or weighting is done across up to 5 trajectories predicted by the baseline models. See Appendix E.3 for detailed explanations of the metrics.

**Baselines** We consider two variants of Robust Imitative Planning (RIP) [72] as baselines. We use an ensemble of probabilistic models to stochastically generate multiple predictions for a given prediction request. Predictions are aggregated across ensemble members via a model averaging (MA) approach. We consider a simple RNN-based behavioral cloning network (RIP-BC) [73] and autoregressive flow–based Deep Imitative Model (RIP-DIM) [74] as backbone models. We adapt RIP to produce uncertainty estimates at two levels of granularity: per-trajectory and per–prediction request. Finally, we vary the number of ensemble members $K \in \{1, 3, 5\}$ and the uncertainty estimation method between Deep Ensembles [14] and Dropout Ensembles [13, 75]. See Appendix E for details on RIP, uncertainty estimation methods, backbone models, experimental setup, and full results. Additional results using Dropout Ensembles are provided in Appendix E.5.

Table 6: Predictive performance of BC & DIM RIP on in-domain, shifted, and full `dev` & `eval` data.

| Dataset | Model | cNLL ↓ | | | minADE ↓ | | | weightedADE ↓ | | | minFDE ↓ | | | weightedFDE ↓ | | |
|---|---|---|---|---|---|---|---|---|---|---|---|---|---|---|---|---|
| | | In | Shifted | Full | In | Shifted | Full | In | Shifted | Full | In | Shifted | Full | In | Shifted | Full |
| Dev | BC, MA, K=1 | 59.64 | 98.54 | 64.29 | 0.818 | 0.960 | 0.835 | 1.088 | 1.245 | 1.107 | 1.718 | 2.113 | 1.765 | 2.368 | 2.777 | 2.417 |
| | BC, MA, K=5 | 56.86 | 91.54 | 61.01 | 0.765 | 0.887 | 0.779 | **1.012** | **1.133** | **1.026** | 1.617 | 1.976 | 1.660 | **2.210** | **2.551** | **2.251** |
| | DIM, MA, K=1 | **50.66** | 73.00 | **53.34** | 0.750 | 0.818 | 0.758 | 1.523 | 1.583 | 1.530 | 1.497 | 1.720 | 1.524 | 3.472 | 3.639 | 3.492 |
| | DIM, MA, K=5 | 50.85 | **72.45** | 53.43 | **0.719** | **0.786** | **0.727** | 1.399 | 1.469 | 1.408 | **1.482** | **1.698** | **1.508** | 3.202 | 3.393 | 3.225 |
| Eval | BC, MA, K=1 | 60.20 | 98.82 | 67.93 | 0.829 | 1.084 | 0.880 | 1.104 | 1.407 | 1.164 | 1.733 | 2.420 | 1.870 | 2.394 | 3.197 | 2.555 |
| | BC, MA, K=5 | 57.75 | 95.00 | 65.20 | 0.777 | 1.014 | 0.824 | **1.028** | **1.299** | **1.082** | 1.636 | 2.278 | 1.765 | **2.238** | **2.957** | **2.382** |
| | DIM, MA, K=1 | **50.50** | **76.00** | **55.60** | 0.759 | 0.942 | 0.796 | 1.551 | 1.883 | 1.618 | 1.511 | **1.983** | 1.605 | 3.536 | 4.376 | 3.704 |
| | DIM, MA, K=5 | 51.19 | 78.85 | 56.73 | **0.728** | **0.918** | **0.766** | 1.424 | 1.754 | 1.490 | **1.493** | 2.000 | **1.595** | 3.256 | 4.093 | 3.424 |

Predictive performance results for the RIP variants are presented in Table 6. Performance is assessed on the in-distribution (In), distributionally shifted (Shifted), and combined (Full) `dev` and `eval` datasets. We observe that across all model configurations, performance on the shifted data is worse than that on the in-distribution data. We also observe that RIP-BC consistently outperforms RIP-DIM on the per-trajectory confidence weighted metrics (weightedADE and weightedFDE), and RIP (DIM) outperforms RIP (BC) on minADE and minFDE. This result might occur if DIM has higher predictive variance. In such a case, DIM might be more effective in modeling multimodality, and therefore

---

[7]This partitioning is also the one used in the Shifts Challenge: http://research.yandex.com/shifts

would tend to produce at least one high accuracy trajectory on more scenes, improving performance on `min` aggregation metrics. This is supported by DIM models yielding the best cNLL, which is a metric particularly sensitive to correct treatment of multi-modal situations. In contrast, for "obvious" scenes, DIM might then produce unnecessarily complicated trajectories which would be reflected in poor performance on weightedADE.

Table 7: Uncertainty and robustness performance for Vehicle Motion Prediction. The error metric for computing the area under the F1 curve (F1-AUC) and F1 at 95% retention rate (F1@95%) is **cNLL**.

| Data | Ensemble Size (K) | R-AUC cNLL ↓ | | R-AUC weightedADE ↓ | | F1-AUC (%) ↑ | | F1@95% ↑ | | ROC-AUC (%) ↑ | |
|---|---|---|---|---|---|---|---|---|---|---|---|
| | | RIP-BC | RIP-DIM | RIP-BC | RIP-DIM | RIP-BC | RIP-DIM | RIP-BC | RIP-DIM | RIP-BC | RIP-DIM |
| Dev | 1 | 11.22 | 12.86 | 0.268 | 0.419 | 65.1 | 63.8 | 89.3 | 87.4 | 51.0 | **51.8** |
| | 5 | **9.08** | 13.24 | **0.236** | 0.376 | **65.2** | 63.7 | **90.6** | 89.7 | 49.2 | 51.4 |
| Eval | 1 | 12.91 | 14.32 | 0.293 | 0.458 | 65.0 | 63.6 | 88.4 | 86.3 | **52.8** | 51.8 |
| | 5 | **10.57** | 15.16 | **0.258** | 0.411 | **65.1** | 63.5 | **89.7** | 88.9 | 52.1 | 50.9 |

Table 7 presents a joint evaluation of the uncertainty quantification and robustness of our baselines. We compute R-AUC with respect to cNLL and weightedADE, and the F1-AUC and F1@95% metrics with respect to the cNLL metric, as detailed in Appendices A and E.3. We observe that an ensemble of RIP-BC models outperforms RIP-DIM on these metrics. These results strongly suggest that RIP-BC has more informative uncertainty estimates than RIP-DIM, because RIP-BC achieves better R-AUC cNLL despite having greater overall error in terms of cNLL (in addition to minADE and minFDE). Figure 3 depicts, for cNLL, error- and F1-retention curves on the full `eval` dataset which reflect the trends observed in Table 7. Additionally, we find that across model configurations the per–prediction request uncertainty scores do not perform particularly well in detecting distribution shift (ROC-AUC). This may occur due to significant data uncertainty in all cases. Future work on detecting distributional shift on this dataset could, for example, inspect the distribution of log-likelihood scores on the in-distribution and shifted partitions in order to devise a metric for this task, aside from the uncertainty scores $U$ used for the retention analysis.

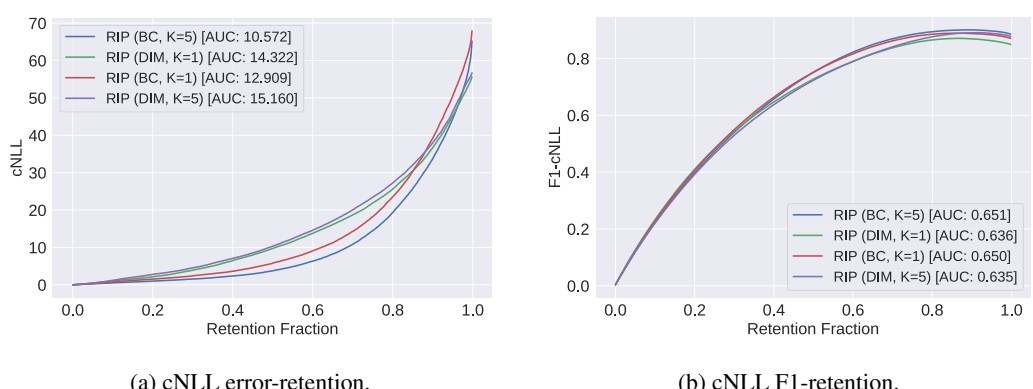

(a) cNLL error-retention.    (b) cNLL F1-retention.

Figure 3: Retention curves for Vehicle Motion Prediction on full `eval` data.

## 6 Conclusion

In this paper, we proposed the **Shifts Dataset**: a large, standardized dataset for evaluation of uncertainty estimates and robustness to realistic, curated distributional shift. The dataset — sourced from industrial services — is composed of three tasks, with each corresponding to a particular data modality: *tabular weather prediction*, *machine translation*, and self-driving car (SDC) *vehicle motion prediction*. This paper describes this data and provides baseline results using ensemble methods. Given the current state of the field, where most methods are developed on small-scale classification tasks, we aim to draw the attention of the community to the evaluation of uncertainty estimation and robustness to distributional shift on large-scale industrial tasks across multiple modalities. We believe this work is a necessary step towards meaningful evaluation of uncertainty quantification methods, and hope for it to accelerate the development of this area and safe ML in general.

## Acknowledgments and Disclosure of Funding

We would like to thank Yandex for providing the data and resources necessary in benchmark creation. We thank Intel and the Turing Institute for funding the work of the OATML Group on this project. Finally, we thank Cambridge University Press and Cambridge Assessment for funding the work of the CUED Speech Group.

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
