# OpenReview forum: "Shifts: A Dataset of Real Distributional Shift Across Multiple Large-Scale Tasks"
_NeurIPS.cc/2021/Track/Datasets_and_Benchmarks/Round2 — NeurIPS 2021 Datasets and Benchmarks Track (Round 2)_

### Official Review · Reviewer_X8AK · 2021-09-15
**A realistic in the wild dataset with diverse tasks**

**Rating:** 7
**Confidence:** 4
**Correctness:** The dataset is constructed in a sound…

**Strengths:**

- I very much appreciate considering the correlation between uncertainty measures and robustness. This is something that has been under-looked.
- Diverse and realistic datasets with real world challenges and distribution shifts.
- many evaluation metrics have been considered.

**Weaknesses:**

- While there are a lot of details in the supplementary materials, the data splits (dev, eval, dev-in, dev-out, eval-in, eval-out) are not well explained in the main manuscript. I think these splits should be better explained.
- I appreciate the authors consider a multiple assessment metrics, however, the datasets themselves are not discussed thoroughly as they should have been in the main script. For example what makes in-domain and out-domain is not described for each dataset in the main text. I had to find those explanations in the supplementary material.
- For each dataset only one baseline and in variants of single and ensemble is presented. I think more baselines and a discussion on the performance of these baselines is needed.

**Additional Feedback:**

All feedbacks have been discussed above.

**Clarity:**

The paper is generally well written and easy to understand. I believe some parts need to be made more clear which I have described above.

**Documentation:**

The documentation for each dataset is described in detail in the supplementary material.

**Ethics:**

There could be some ethical concerns for example in the case of the Reddit dataset, the original owners of the comments could be potentially identified. I recommend an ethical review of the manuscript.


**Relation To Prior Work:**

Yes. The paper mentions that in the presented work the focus is on evaluation of the correlation between uncertainty measure and robustness as opposed to those measures individually.

**Summary And Contributions:**

The paper presents an "in-the-wild" benchmark for evaluating uncertainty and robustness to distribution shift.  The benchmark contains realistic tasks and distribution shifts taken from industrial sources. The benchmark has three parts each part relating to a specific modality: Tabular data, machine translation and self-driving cars.
Instead of looking at robustness and uncertainty measures independently, the paper considers a general paradigm that " uncertainty estimates should correlate with the degree of distributional shift.

---

> ### Author Response · Authors · 2021-09-28
> **Individual Response**
>
> Dear Reviewer X8AK,
>
> Thank you for your hard work and your comments!
>
> As discussed in the general response, we are currently working on improving the discussion of the metrics and dataset splits and using the extra page to do so in the main body of the paper.
>
> We will also be providing additional ensemble-based baselines and will expand our discussion of these results.
>
> Best Regards,
> The Authors

---

### Official Review · Reviewer_Exx5 · 2021-09-19
**Great contribution of creating the large-scale distributional shifts dataset. But it lacks further analysis.**

**Rating:** 7
**Confidence:** 3

**Strengths:**

1. The authors release a large-scale dataset for promoting the development of uncertainty and robustness verification of ML models. The data collected from the various application scenarios can support the motivation of this paper, i.e., verify the ML model in the “wild applications”. I think the contribution and impact are enough.


2. The proposed evaluation paradigm jointly assesses the robustness and uncertainty. It alleviates the problems that the previous evaluation methods focus on the specific data partitions instead of examining the correct relationship between error and uncertainty estimation.


**Weaknesses:**

1. The authors did not provide the comparison of the related works. What are the differences in the data characteristics between *Shifts Dataset.* and other data sets?


2. The authors did not give some preliminary analysis of each task. It is unclear how the distributional shift is truly examined by using *Shifts Dataset*.


3. The authors did not evaluate the different methods of overcoming the distributional shifts using *Shifts Dataset*. Only providing the results of ensemble baselines may not be sound, because this kind of dataset should also distinguish the best/better methods that alleviate the distributional shifts.


4. It is better to provide the human evaluation results for text generation tasks, especially for proposing the new benchmark or evaluation metrics for machine translation. It can help examine the correlation results between human ratings and automated evaluation results.

**Additional Feedback:**

I have some questions about the Section.4 Machine Translation.


1. Do you have plans to support more languages? But I fully understand that collecting the professional translation may be expensive (as depicted in the Appendix), and it is hard to support other languages.

2. Since the pre-trained model is the most popular paradigm in the current NLP field, do you think *Shifts Dataset* can be used to examine the robustness of pre-trained language models?

3. BLEU cannot evaluate the semantic-level features. Why not introduce more evaluation metrics that correlate well with human judgements, e.g., COMET, BLEURT, BERTScore?

The PDF should not be generated with preprint option. There are typos in this paper. For example, check them:

- Abstract & Introduction: In this work we -> In this work, we
- Section.3.Baselines: appendix -> Appendix
- Section.5.Dataset: appendix -> Appendix
- Section.5.Baselines: appendix -> Appendix

**Clarity:**

The paper is well written with clear logic. The author has claimed the motivation and the evaluation metric. Then the application results can support the effectiveness of this dataset. But I found some typos in this paper, see the “Additional Feedback” part.

**Correctness:**

The construction of the dataset and the evaluation method look sound. But the caption of Figure 2 is obviously wrong. And I am also curious about why the two curves shown in Figure 2 are very close. Does it mean that the proposed evaluation metric cannot work well under this situation?

**Documentation:**

The codes are available with clear documentation. The authors provided detailed maintenance and licensing plans in the Appendix.


**Ethics:**

The data was collected from real-world application scenarios. I do not see any ethical problems except for Reddit data. Are there any aggressive opinions regarding extreme political comments, racial problems, etc.?


**Relation To Prior Work:**

The paper adequately cited the prior work. For other concerns regarding the previous work, please refer to the “Weakness” part.

**Summary And Contributions:**

The paper presents a series of datasets called *Shifts Dataset*, aiming to evaluate the ML models’ ability in terms of uncertainty estimation and robustness. The main idea is to construct the dataset by shifting the test data from the distribution of training/in-domain. Specifically, the data sources come from the industry services and consist of three tasks in order to fit the real-world distribution including multiple modalities: tabular, text and computer vision. For evaluation of distributional shift, the authors also propose a more general evaluation paradigm considering the correlation between uncertainty estimation and the degree of error. For verifying the effectiveness/correctness of the collected data, the authors also conducted the experiments on *Shifts Dataset* using the trained ensemble baseline models. The results evaluated by the error-retention curves show that the datasets are reasonable.

---

> ### Author Response · Authors · 2021-09-25
> **Some questions**
>
> Hello!
>
> We're currently addressing most of your and your colleagues comments and will soon post general and point-by-point replies as well as provide an updated manuscript. However, we're not sure we fully understand points 1-3 in your "weaknesses section". For us to be better able to address your concerns, could you please clarify the following:
>
> 1. What kind of comparison and to which related works are you referring? Other robustness and uncertainty datasets, such as WILDS and ImageNet A/C/R/O ? Or different algorithms?
>
> 2. What kind of preliminary analysis did you have in mind? Upon examination of the WILDS dataset, for example, we didn't see preliminary analysis (excluding baseline numbers) which were more substantial than what we provide in the main and appendices. Could you please concrete examples of analysis, so that we could potentially include them.
>
> 3. What kind of baselines are you referring to? Things like Invariant Risk Minimization?
>
> Best Regards,
> Authors

---

> > ### Comment · Reviewer_Exx5 · 2021-09-25
> > **Reply to the authors' questions**
> >
> > Hi authors,
> >
> > Sorry for any inconvenience caused. I will clarify here:
> >
> > 1. Yes. I refer to the benchmark comparison, like the ImageNet A/C/R/O you mentioned above. But this paper did not talk about the image classification task, I think other popular out-of-domain/corrupted data that are used to examine the distributional shifts can be viewed as comparable benchmarks.
> >
> > 2. For example, when we talked about the textual dataset, we may compare the distribution of word frequency, the constitution of part-of-speech, etc. They are helpful to analyse the differences among various benchmarks. I am interested in why the *Shifts Datasets* is better than other benchmarks from the perspective of data characteristics.
> >
> > 3. Yes, things like Invariant Risk Minimization. The paper only considered the model ensemble methods.  I am interested in how the *Shifts Dataset* can verify other algorithms that are used to overcome the distributional shifts, such as domain adaptation.
> >
> > Best regards,
> > Reviewer Exx5

---

> > > ### Author Response · Authors · 2021-09-25
> > > **Reply**
> > >
> > > Thanks for the fast reply!
> > >
> > > Regarding (1) - Does the discussion in the 3-5th paragraphs of the introduction (page 2) ("While much work.... ....using ensemble methods") not sufficiently answer the question about how Shifts is qualitatively different from prior benchmarks and what it adds? Specifically, that most (but not all) prior benchmarks focus on small--mid-scale image classification and consider synthetic shifts. The world isn't limited to images as a modality or classification as a task and synthetic shifts may not be representative of real-world shifts. Shifts specifically tries to answer this need for large scale datasets across multiple modalities which consider different tasks and real-world shifts.
> > >
> > > If the discussion in the intro is insufficient we are happy to expand upon it. :)
> > >
> > > Best,
> > > Authors

---

> > > > ### Comment · Reviewer_Exx5 · 2021-09-25
> > > > **Reply to authors**
> > > >
> > > > Thanks for your detailed explanation.
> > > >
> > > > I agree that the real-world data is more reliable than the synthetic/corrupted data. But you know that, the experimental results are more credible. I think the qualitative discussion is fine for me. Take it easy! I just pointed out a minor weakness regarding the quantitative comparison. I did not meant to force you to do these experiments due to the workload and page limitation.
> > > >
> > > > Best regards,
> > > > Reviewer Exx5

---

> ### Author Response · Authors · 2021-09-28
> **Individual Response**
>
> Response:
>
> Dear Reviewer Exx5,
>
> Thank you for your hard work and your comments! Please let us address your comments.
>
> ### Weaknesses
>
> 1. We discussed the relationship on Shifts to prior benchmarks a little earlier already.
> 2. We are not certain  how analysis of word distributions, for example, will be helpful, and it’s unclear how to represent that diagrammatically, as the NMT models have BPE-token vocabularies of over 40000 tokens. However, we _do_ demonstrate that on shifted data all models in all datasets suffer a clear drop in performance. However, we will add additional dataset statistics to tabular and nmt sections. Specifically, precipitation class distributions and analysis of sentence lengths.
> 3. As discussed in the general response, robust learning approaches such as IRM require domain-annotation at training time and do not provide any uncertainty estimates. Thus, they don’t quite fit into our evaluation paradigm, where we assign equal importance to robustness and uncertainty and assume no information about possible shifts at training or test time (which we admit is a rather extreme scenario). However, we will provide additional ensemble-based baselines of differing quality to demonstrate how the datasets allow distinguishing between various approaches for uncertainty estimation popular in the Bayesian Deep Learning literature.
> 4. While we agree with you in general, in this context human evaluation is not particularly appropriate for one crucial reason – humans are exposed to far more knowledge than our models, and it would not be a valid comparison of robustness and uncertainty estimation between models trained on a particular dataset versus humans with the vastness of their linguistic knowledge.
>
> ### Additional Feedback
>
> 1. Yes, we have plans to eventually add additional languages and examine alternative sources of informal language. However, production of these datasets is far slower than the weather or SDC data, as we need human translation in a range of languages and also human annotations of anomalies.
> 2. This is a really great question! This really touches upon the interaction of transfer learning, distributional shift, and robustness. I think an evaluation would be possible if we were able to strictly know what data models were pre-trained on and to which knowledge they have been exposed, in order to design appropriate downstream distributionally shifted datasets. Otherwise, it might not be a clean experiment. This is very related to the important issue of test leakage in large language models.
> 3. BLEURT, BERTScore and related metrics look very promising for assessment of quality of language generation. However, they aren’t entirely easy to use or appropriate in the context of our setup. Firstly, they require access to a pre-trained language model, which may not be available (though many are for En-Ru) and whose training data may be poorly matched to what we evaluate on. Secondly, some metrics (BLEURT and similar) examples of human evaluation for training and fine-tuning - we don’t have such examples available. Thirdly, and more importantly, these metrics are known to be very sensitive to distributional shift and may yield poor scores when exposed to shifted data, which is exactly the scenario we are examining. Thus, while BLEU (and GLEU) has known deficiencies, it is nevertheless robust to distributional shift. That said, the development of metrics for assessing robustness and uncertainty has lagged behind methods and datasets. This is exactly why in the Shifts Challenge at NeurIPS2021, which is organized around the Shifts Datasets, we explicitly invite participants to submit extended abstracts on evaluation metrics. In this context, investigation of alternative assessment metrics, including BLEURT, BERTScore,  is a very interesting avenue for future research.
>
>
> Best Regards,
> The Authors

---

> > ### Comment · Reviewer_Exx5 · 2021-09-29
> > **Reply to the authors**
> >
> > For Q1, the authors address my concern.
> >
> > For Q2, I agree with the authors' reply and would be happy to see future work in terms of testing the pre-trained language model. (I'm not forcing the authors to add it to the camera-ready version because it is a new topic.)
> >
> > For Q3, there is no denying that the distributional shift can not affect the evaluation results of BLEU and GLEU. But the paper of BLEURT already talked about the robustness problem regarding the domain shift. Since the author plan to invite participants of the Shifts Challenge to submit extended abstracts on evaluation metrics, I will not challenge this point.
> >
> > Thanks for the reply. I would like to raise my rating of this paper.

---

### Official Review · Reviewer_yHpU · 2021-09-19
**Review of Shifts: A Dataset of Real Distributional Shift Across Multiple Large-Scale Tasks**

**Rating:** 5
**Confidence:** 3

**Strengths:**

I agree with the authors on the necessity of a large scale dataset in the literature for evaluation of uncertainty estimates and robustness to distributional shift.
Some sections of this paper are very well-written, while some are surprisingly explained very poorly. The data collection process is explained in detail and the data descriptions are very clear.


**Weaknesses:**

## Major Comments:
1) I think literature on domain adaptation/transfer learning/multi-task learning is missing in the current related work review. Including the related literature on these fields would improve first accessibility of the paper and second would make it much easier for researchers in different fields to relate.
2) I am not sure about the ensemble-based approach to uncertainty. What does it mean to train ensemble of ten models on the train data? Please provide more explains on "Single" and "Ens" that appears in Table 1, Table 2, and Table 4.
3) How does the distributional shift in the weather data for example relate to concept drift?
4) What does a normal laptop mean? Please provide a description for it.
5) I do not understand assessment metrics explained in Appendix A. This section in particular is very badly written and it has several grammar mistakes. I cannot indicate all of the mistakes because the submitted template does not have the line numbers.

## Minor Comments:
1) This paper is submitted in wrong template. The correct template is NeurIPS 2021 Datasets and Benchmarks Track template that is available in https://neurips.cc/Conferences/2021/CallForDatasetsBenchmarks.
2) Page 4-Line 9: “A” should be capital in appendix C.4.
3) Section 4-Dataset-1st Line- "(eval)" instead of "(eva)".
4) Note that the font style of eval/dev is not consistent throughout the paper. Is it on purpose?
5) Typo in Appendix A: 'This represents a a hybrid hu...'.
6) Figure labels are not matching in font sizes. For example see, Figure 5 (a) and Figure 5 (b).

**Additional Feedback:**

Please see my comments above.

**Clarity:**

The data collection process is clearly explained in detail.
Unfortunately, I do not think the evaluation metrics are explained well. Especially, error-retention curves are not explained clearly. To understand the concept one should read [11] and even there the concept of error-retention curves does not appear directly. I would suggest the authors to directly indicate the definition of error-retention curves in the reference or explain it in detail within the paper for the sake of completeness.

**Correctness:**

Based on my understanding the data is constructed in a sound way. However, the evaluation methods are not clearly explained in the paper. Therefore, I am not sure if performances of the baseline models are evaluated reasonably.

**Documentation:**

The dataset collection process is very well-documented.

**Ethics:**

No.

**Relation To Prior Work:**

I think literature on domain adaptation/transfer learning/multi-task learning is missing in the current related work review. Including the related literature on these fields would improve first accessibility of the paper and second would make it much easier for researchers in different fields to relate to.


**Summary And Contributions:**

This paper propose a dataset for evaluation of uncertainty estimates and robustness to naturally curated distributional shift. The dataset is collected from an industrial source for three different tasks, namely tabular weather prediction, machine translation, and self-driving car (SDC) vehicle motion prediction.
All tasks are affected by real distributional shifts.
A description of the dataset and the baseline results for all tasks are provided.

---

> ### Author Response · Authors · 2021-09-28
> **Individual Comments**
>
> Dear Reviewer yHpU,
>
> Thank you for your hard work and your comments!  Please allow us to  address your feedback.
>
> ### Major Comments
>
> 1. Regarding the related literature – we have provided a discussion on the comparison to domain-adaptation and robust learning in the general response. We will also significantly extend our discussion of related work with domain-adaptation and robust learning literature, in order to make the paper more accessible to readers from backgrounds other than Bayesian Deep Learning. If you have particular citations which we ought to cover in this review, we would greatly appreciate your recommendation!
>
> 2. Regarding ensemble baselines – ensemble approaches, such as Deep Ensembles [2,5,6,7,8,9,10 in general response] are the de-facto standard method for uncertainty estimation in the Bayesian Deep Learning community. They are known to provide interpretable uncertainty estimation, yield improved predictive performance, can be applied to many tasks without significant adaptation, and do not require domain annotations at training time. As discussed in the general response, ensembles improve robustness by averaging over different models, each of which may have different spurious correlations, such that the average prediction is more representative of the systematic correlations in the data.
>
> 3. Thank you for this insightful point – the time-shift in weather can indeed be seen as concept  drift, and we will reference this in our discussion.
>
> 4. We apologize for not being clear enough on this point. By a normal laptop we meant a modern mid-range laptop without a dedicated GPU: for example, a laptop with an Intel i5 quad core processor and 8 GB of RAM. We will clarify this in the text.
>
> 5. As discussed in the general response, we are working on an updated manuscript where we improve the clarity of discussion of metrics and datasets. We will upload the new manuscript soon.
>
> ### Minor Comments
>
> Thank you for pointing out these errors - we have already addressed them and will soon upload an improved manuscript.
>
> Best Regards,
> The Authors

---

### Official Review · Reviewer_D27n · 2021-09-21

**Rating:** 6
**Confidence:** 3
**Clarity:** The paper was written clearly.

**Strengths:**

- The benchmark includes large-scale datasets with "in-the-wild" distribution shifts, and it includes various modalities (tabular, text, self-driving car scenes) and tasks (weather prediction, machine translation, and motion prediction). These modalities and tasks are different from what's typically studied in a benchmark for robustness, so I think the benchmark will be a useful resource for the community.
- The paper is clearly written, with a detailed appendix.

**Weaknesses:**

A few of the datasets consider fairly drastic shifts, which makes me wonder whether the distribution is "in-the-wild" as well as tractable:
- The weather prediction dataset studies a distribution shift from tropical, dry, and temperate climates to snow and polar climates, in addition to shift over time. This distribution shift across the climates seems poorly motivated to me (you'd train on all climates typically), and it also seems very challenging for a weather prediction model to generalize to vastly different climates. Could you explain? In addition, do you have a sense of how much of the difference in performance between eval-in and eval-out is explained by the changes over climates, as opposed to changes over time?
- The translation dataset studies generalizing from "standard" language to atypical language such as slangs. I understand the general motivation, but I have concerns about the tractability of generalizing to atypical language from just standard language.

Baseline algorithms for improving robustness would be nice to benchmark. The current paper only reports performance of standard models.

#### Minor comments
In the first paragraph of section 3, which introduces the tabular weather prediction dataset, the authors seem to claim that the proposed dataset is appropriate for measuring robustness on tabular data in general, rather than just specifically for the distribution shift, task, and data studied on the dataset, by arguing that there are similar distribution shifts in other tabular data such as financial and medical data. I think this over-claims the applicability of the dataset since it's unclear whether robustness for a certain distribution shift and a dataset transfers to robustness for a different distribution shift and a dataset, even if the modality is the same.

The machine translation dataset is a combination of standard corpora and the MTNT dataset, which is the shifted test set with atypical language. The MTNT paper studies basically this setting and demonstrates that models perform poorly on the MTNT dataset. The contributions for this particular dataset seem somewhat limited.

**Additional Feedback:**

Typo in the Dataset paragraph of Section 4: "evaluation (eva) data"

**Correctness:**

The paper is correct, although there are some parts that could be toned down or be more nuanced as described in other parts of the review.

**Documentation:**

The paper provides detailed documentation on the datasets and includes a URL for accessing the data. Appendix B includes details on distribution, maintenance, and ethical use.

**Ethics:**

Not to my knowledge

**Relation To Prior Work:**

The authors provide a thorough comparison with related work, both on methods and benchmarks. The authors also claim that their benchmark is novel because it considers a range of modalities and tasks, whereas previous benchmarks consider images and classification tasks. The diversity in modality and tasks are definitely a strength of the Shifts dataset as authors point out, but I think the characterization of prior work could use more nuance since existing benchmarks for distribution shifts also consider modalities and tasks beyond image classification. For example, many standard domain adaptation benchmark consider detection tasks.

**Summary And Contributions:**

The authors propose a new benchmark for uncertainty estimates and robustness to "in-the-wild" distribution shifts, called Shifts. The Shifts benchmark includes three datasets from different modalities and tasks, which are weather prediction on tabular data, machine translation, and vehicle motion prediction on self-driving car data.

---

> ### Author Response · Authors · 2021-09-25
> **Some questions**
>
> Hello!
>
> We're currently addressing most of your and your colleagues comments and will soon post general and point-by-point replies as well as provide an updated manuscript. For us to be better able to address your concerns, could you please clarify to which "Baseline algorithms for improving robustness" you are referring to?
>
> Best Regards,
> Authors

---

> ### Author Response · Authors · 2021-09-28
> **Individual Response**
>
> Dear Reviewer D27n,
>
> Thank you for your hard work and your comments!  Please let us address your feedback.
>
> ### Realism of Shifts
> The idea is to study robustness and uncertainty estimation in _equal_ measure. The dev and eval datasets consist of both in-domain and shifted components. The shifted component is designed to feature more drastic forms of shift to which generalization will be difficult – on these examples we want the model to say “I don’t know, I think I’ll get it wrong”. However, not all examples within the shifted set are so extreme, and on those the models can presumably do better.
>
> Regarding the climate shift – we specifically designed this climate shift so that there will be data on which models will find it hard to generalize. However, certain locations will be near to one in which we have data, so on those regions we can generalize better. Similarly for the Reddit data; not all Reddit data features anomalies so dramatic that a model will entirely fail to generalize. In Appendix C we describe the anomaly distribution and co-occurrence of anomalies in the Reddit data. We expect that on data with fewer or less drastic anomalies we will be able to generalize better, and on sentences which feature either more drastic anomalies or greater numbers of co-occurring anomalies, models will find it more challenging to generalize and will be forced to say “I don’t know”.
>
> ### Baselines
> The ensemble-based baselines we consider allow us to achieve improved robustness on shifted data, as well as to calculate per-prediction uncertainty estimates. Using only robust learning baselines such as IRM would ignore half of our evaluation paradigm – uncertainty estimation. However, we agree that our work could benefit from comparisons across uncertainty quantification methods, and will therefore provide additional ensemble-based baselines.
>
>
> ### Minor Comments
>
> We agree that we can moderate this statement. However, what we meant isn’t that the shifts are the same in all tabular tasks; rather, that the challenges are similar. Tabular datasets have features from different sources (e.g., measurement devices), missing values, features of widely different dynamic ranges, mixtures of continuous and categorical variables, dataset drift over time, etc… Thus, studying how to work with tabular weather data should also provide insights into how best to improve robustness and uncertainty estimation on other tabular tasks. We definitely agree that different  tabular datasets will not feature identical shifts.
>
> The original MTNT exclusively studied the robustness aspect, had a significantly smaller evaluation dataset and did not have curated anomaly flags. In our version, we are interested in assessing _both_ a model’s robustness _and_ its ability to indicate inability to translate via uncertainty estimation. Hence, we provide not only additional Reddit data, but also clean news translation data matched to the training set. On this combined dataset, models perform well on most (but not all) news data, and poorly on most (but not all) Reddit data. Ideally, models would  perform better overall in either situation, but in practice, the models should also be able to indicate when they expect to do well or poorly using uncertainty estimates (regardless of whether they are considering news or Reddit data).
>
> Regarding moderating our claims about other benchmarks and tasks, we would greatly appreciate it if you could point out any particular domain adaptation benchmarks which we have missed. We will cite them and compare our work to theirs. Regarding tasks -- to be specific, by task we mean whether a model is trained for e.g., regression, classification, or something else. Most tasks in uncertainty quantification literature focus on classification, rather than regression or structured prediction tasks.
>
> Best Regards,
> The Authors

---

### Author Response · Authors · 2021-09-28
**General Response Part 1**

Dear Reviewers,

Thank you for your hard work and insightful reviews! In addition to this general response, we will also respond to each of you, to address individual comments. We are working on an updated manuscript to be uploaded in a few days, taking into consideration feedback on the presentation of the dataset and metrics, and adding additional baselines.

However, based on your comments we are a little concerned that we are from different backgrounds and that we may view the problems of robustness somewhat differently. Please let us attempt to bridge this gap, by explaining our world view and how we structured the Shifts Dataset around it.

### Paradigm

First, we view the problems of robustness and uncertainty estimation as having _equal_ importance -- models should be robust, but where they are not, they should yield high estimates of uncertainty, which enables risk-mitigating actions to be taken (for example, a self-driving car transferring control to a human operator).

Second, we assume that at training or test time _we do not know a priori_ about alternative domains and whether or how our data is shifted. This setup aims to emulate real-world deployments in which the variation of conditions is vast and one can never collect enough data to cover all situations. It is for this reason we view robustness and uncertainty as equally important -- we assume that one can never be fully robust in all situations, and it is in these situations that high-quality uncertainty estimation is crucial. This is a strictly more challenging setting than one in which auxiliary information about the degree or nature of shift is available at training or test time.

### Dataset Construction

We have constructed the Shifts Dataset within the context of the two above points. Specifically, the dataset is constructed with the following attributes. First, the annotations of distributional shift are meant to be used for analysis rather than model construction.  Second, we have explicitly partitioned the datasets such that the shifts are realistic but significant and will be challenging to be fully robust to -- this allows us to assess the quality of uncertainty estimates. However, the weather and motion prediction datasets _can_ be repartitioned in alternative ways which are different from our canonical partitioning, such that alternative robustness paradigms can be evaluated (e.g., by constructing a training set with multiple training distributions, and assuming that models may access the corresponding domain labels at training time). Tools for partitioning and repartitioning are provided in our GitHub repo [1].

### Choice of Baseline Methods

In our work we have exclusively used ensemble-based baselines. This was done for a number of reasons. Firstly, ensemble-based approaches are a standard way to obtain _both_ improved robustness _and_ interpretable uncertainty estimates.  Secondly, ensemble methods are straightforward to apply to any task of choice and require very little adaptation. In the Bayesian Deep Learning literature, it is well-established that Deep Ensembles [2,6,7,8] are a very strong baseline for improving both robustness and uncertainty estimation. Ensembles improve robustness because each model represents a different hypothesis about how the world works. When models are combined, we are effectively marginalizing over different explanations of the data. Conceptually, even if each individual model in an ensemble is subject to spurious correlations, the models will have different spurious correlations. Thus, when the models are combined, the effects of spurious correlations are cancelled out to a certain degree, improving generalization performance. Uncertainty estimates can be obtained from measures of ensemble diversity -- if the predictions are highly diverse, then the ensemble members cannot agree on what the prediction should be and therefore are highly uncertain.

Other than ensemble methods [2,5,6,7,8,9,10], there are very few alternative approaches which are known to yield improved robustness _and_ interpretable uncertainty estimates, can be easily applied to a broad range of large-scale tasks without significant adaptation of the method, and do not require extra information about the nature of distributional shift at training or test time.

---

> ### Author Response · Authors · 2021-09-28
> **General Response Part 2**
>
> ### Comparison to Domain Adaptation or IRM Methods
>
> In domain adaptation literature, the paradigm is different to the one described above. The aim is to adapt a model to an explicitly known target domain, whose attributes are either partially or completely known [3]. Furthermore, uncertainty estimation is not a focus of this context.
>
> In the context of robust learning, IRM [4] and related methods require domain annotations at training time so that models can be explicitly constructed to avoid learning domain-specific spurious correlations and instead learn the true underlying interactions. Again, uncertainty estimation rarely appears in this context.
>
> While the Shifts Dataset was not constructed with these applications or evaluation paradigms in mind, the weather and motion prediction datasets _can_ be re-partitioned to be used in these contexts, making use of domain annotations over, for example, climates or locations, respectively. We leave investigation of domain adaptation methods on another partitioning of the Shifts Dataset to future work.
>
> [1] https://github.com/yandex-research/shifts
>
> [2]  Lakshminarayanan et al., “​​Simple and Scalable Predictive Uncertainty Estimation using Deep Ensembles”, 2017.
>
> [3]   Mei Wang, Weihong Deng, “Deep Visual Domain Adaptation: A Survey”, 2018.
>
> [4] Martin Arjovsky, Léon Bottou, Ishaan Gulrajani, David Lopez-Paz. “Invariant Risk Minimization”, 2019.
>
> [5]  Ovadia, et al., “Can You Trust Your Model's Uncertainty? Evaluating Predictive Uncertainty Under Dataset Shift”, 2019
>
> [6] Ashukha et al., “Pitfalls of In-Domain Uncertainty Estimation and Ensembling in Deep Learning”, 2020
>
> [7] Filos, Angelos and Tigkas, Panagiotis and McAllister, Rowan and Rhinehart, Nicholas and Levine, Sergey and Gal, Yarin, “Can autonomous vehicles identify, recover from, and adapt to distribution shifts”, 2020
>
> [8]  Malinin, Andrey and Gales, Mark, “Uncertainty Estimation in Autoregressive Structured”, 2021
>
> [9] Yarin Gal and Zoubin Ghahramani, “Dropout as a Bayesian Approximation: Representing Model Uncertainty in Deep Learning”, 2016
>
> [10] Maddox, Wesley and Garipov, Timur and Izmailov, Pavel and Vetrov, Dmitry and Wilson, Andrew Gordon, “A simple baseline for bayesian uncertainty in deep learning”, 2019

---

### Author Response · Authors · 2021-09-30
**Manuscript Update**

Dear Reviewers!

We have now uploaded a revised manuscript, where we hope we have fully addressed your concerns regarding clarity, dataset descriptions, and baselines. The modifications are as follows.

### Related Work and Paradigm Discussion

We have significantly expanded Section 2, where we detail our paradigm, and how it informs both dataset construction and choice of baselines.

### Dataset Descriptions

We have updated and expanded the dataset descriptions for tabular and NMT data in Sections 3 and 4 of the main paper. We have also provided a description of the precipitation class distribution for tabular data in Appendix C and sentence-length details for NMT in Appendix D.

### Metrics Descriptions

We have significantly revised Appendix A with clear and formal definitions of  Error-Retention and F1-Retention curves.

### Additional Baselines

We have provided additional ensemble-based baselines for tabular data (in Appendix C.5), where we considered Deep Ensembles [1] and Monte-Carlo Dropout Ensembles [2] of FT-Transformer neural network models. We have similarly provided additional results for the Motion Prediction task in Appendix E.5. Unfortunately, we were not able to provide checkpoint-ensemble baselines for Machine Translation, as our intermediate checkpoints were automatically deleted during data cleaning on the system. We are currently re-training those models and will add additional baselines into the Machine Translation section as soon as they are available.

### Updated Metric for Motion Prediction

Prior to the rebuttal, the Yandex SDC team discovered that weightedADE is not entirely appropriate for assessing performance in situations with inherent multi-modality in Vehicle Motion Prediction, such as T-junctions. Under weightedADE, an optimal model would predict a single _median_ trajectory going straight, rather than predict two different modes, with one going left and one going right. We have introduced a new error metric for the Motion Prediction task, called corrected Negative Log-Likelihood (cNLL), which overcomes this issue. This metric is detailed in Appendix E.3.  Under cNLL, an optimal model would predict two distinct modes and appropriately weight them. We have replaced joint-assessment results for Motion Prediction—which were previously computed using weightedADE—with results computed using cNLL. To our knowledge, we are the first to notice such an issue with weightedADE and propose an alternative metric which overcomes its issues.


### Minor Modifications

We have used the correct template and improved the consistency of all formatting and captions.


We would like to thank the reviewers for their feedback which helped us to improve the paper! We sincerely hope that our responses, the updated manuscript, and additional baseline experiments have sufficiently addressed your concerns regarding our paper. Please feel free to reach out with any additional questions you might have!

Best Regards,
The Authors

–

[1] Lakshminarayanan et. al, Simple and Scalable Predictive Uncertainty Estimation using Deep Ensembles, 2017.

[2] Smith and Gal, Understanding Measures of Uncertainty for Adversarial Example Detection, 2018.

---

### Author Response · Authors · 2021-10-05
**We are happy to address any remaining concerns**

Dear Reviewers D27n, yHpU and X8AK,

We would like to thank you for your feedback - it has helped us to improve the paper! We've done our best to address your concerns in our replies and in the updated manuscript. However, we are uncertain whether we have successfully resolved all of your concerns. If you have any remaining concerns, we are more than happy to engage with you right up to the end of the rebuttal stage! :)

Best Regards,
Authors

---

### Decision · Program_Chairs · 2021-10-09

**Decision:**

Accept

**Comment:**

The paper presents a welcome addition of distribution shifts beyond the image classification domain. The reviewers are generally favorable of the paper, hence I recommend acceptance.

For a more comprehensive overview of robustness research in the image domain, I recommend the authors refer the reader to the following robustness evaluations: https://arxiv.org/abs/2007.01434 , https://arxiv.org/abs/2007.00644 , and https://arxiv.org/abs/2007.08558